# Silk Fibroin Microneedles for Transdermal Drug Delivery: Where Do We Stand and How Far Can We Proceed?

**DOI:** 10.3390/pharmaceutics15020355

**Published:** 2023-01-20

**Authors:** Zhenzhen Qi, Zheng Yan, Guohongfang Tan, Tianshuo Jia, Yiyu Geng, Huiyan Shao, Subhas C. Kundu, Shenzhou Lu

**Affiliations:** 1National Engineering Laboratory for Modern Silk, College of Textile and Clothing Engineering, Soochow University, Suzhou 215123, China; 23Bs Research Group, I3Bs Research Institute on Biomaterials, Biodegrabilities, and Biomimetics, Headquarters of the European Institute of Excellence on Tissue Engineering and Regenerative Medicine, University of Minho, AvePark, Guimaraes, 4805-017 Barco, Portugal

**Keywords:** silk fibroin microneedles, transdermal delivery, intelligently responsive, clinical transformation

## Abstract

Microneedles are a patient-friendly technique for delivering drugs to the site of action in place of traditional oral and injectable administration. Silk fibroin represents an interesting polymeric biomaterial because of its mechanical properties, thermal stability, biocompatibility and possibility of control via genetic engineering. This review focuses on the critical research progress of silk fibroin microneedles since their inception, analyzes in detail the structure and properties of silk fibroin, the types of silk fibroin microneedles, drug delivery applications and clinical trials, and summarizes the future development trend in this field. It also proposes the future research direction of silk fibroin microneedles, including increasing drug loading doses and enriching drug loading types as well as exploring silk fibroin microneedles with stimulation-responsive drug release functions. The safety and effectiveness of silk fibroin microneedles should be further verified in clinical trials at different stages.

## 1. Introduction

Skin is the first line of defense against infection and injury to protect the human body from harmful environments [1,2]; it plays a pivotal role in maintaining essential functions of host physiology. According to rough estimates, skin covers an area of about 1.5–2.0 m^2^ in the human body [3], accounting for nearly one-fifth of the total weight of people, making it the largest organ in the human body. Skin comprises the epidermis, dermis and subcutaneous tissue [4]. As the outermost layer of skin, the epidermis is generally 100–240 μm [3,5]. Local and systemic drug administration with the skin as the primary organ can be found in the oldest surviving medical records [6], now known as transdermal drug delivery. As early as 3000 BC, descriptions of ancient Babylon note ointments and perfumes containing plant and mineral extracts to treat diseases [6]. The key to transdermal drug delivery is to understand the skin’s structure, function and composition, which is also a crucial point in the design of transdermal drug delivery-related drugs and the form of administration [7].

Drug penetration through the skin can be divided into three main types: intercellular, intracellular and follicular [8]. Small molecule drugs are transported to the continuous subcutaneous tissue through the intercellular pathway. The polar/non-polar solute assigned by the appropriate O/W coefficient can diffuse the drug into water and keratinocytes via intracellular pathways [5,9,10] but with reduced bioavailability. The most primitive transdermal drug delivery is a patch, in which the required drug is used as a paste, wrapped with bandages or gauze, and then applied to the affected area for local treatment [11]. The drugs suitable for this type have small molecular weight and low availability [12,13], which leads to the large number of drugs needed for treatment and the possibility of excessive waste. To improve drug availability and permeability, different methods are used to improve skin permeability. Commonly used methods include chemical substances, iontophoresis, ultrasound, etc. [14,15]. However, these methods commonly cause irreversible physiological damage to human tissues, and repeated use in the same place may cause skin ulceration; thus, they are not suitable for long-term use. The latest and most used methods include new chemical enhancers, electroporation, thermal ablation, microcrystalline rejuvenation and microneedles. These technologies enable macromolecular drugs and biological drugs to better penetrate the cuticle to the dermis, improve the efficiency of transdermal delivery, and enhance drug availability. However, the use of additional chemical reagents as well as lasers and heat will cause certain damage to the skin and can cause discomfort for patients [16].

Microneedles are a new transdermal method obtained by combining a transdermal patch with a hypodermic needle [17]. The length of the microneedle can be controlled to easily puncture the cuticle of the skin without touching the dermis and subcutaneous tissues with pain nerves, and the drug can be released without pain [18], thus achieving high drug availability. In addition, microneedles can be easily removed and provide better drug slow-release function, further reducing patients’ dependence on drugs [19,20,21]. At the same time, microneedles, as drug carriers, are extremely helpful for releasing macromolecular medications [22,23].

The synthetic polymer material used as microneedles’ substrate has the problems of a complex preparation process and poor biocompatibility [24]. Collagen, hyaluronic acid, polysaccharide, silk fibroin and other natural polymer materials have good biosafety and are a source of environmental protection. However, due to the mechanical properties, the application of collagen, hyaluronic acid and polysaccharide in microneedles is limited. Silk fibroin has highly elastic mechanical properties [25], a significant raw material source, and many unique advantages, such as good biological stability of enzymes [26,27], antioxidant activity stability [28,29], and biological stability in biomolecular drugs [30] and vaccines [31,32]. Silk fibroin is often prepared into films, hydrogels, nanoparticles and microneedles, which can be used in various fields such as medical cosmetology, tissue engineering and drug delivery. In particular, the preparation of microneedles plays a pivotal role in drug delivery [18,33,34,35]. This review focuses on the latest research progress of silk fibroin in the field of microneedles, analyzes in detail the structure and properties of silk fibroins, the types of silk fibroin microneedles, drug delivery applications and the clinical transformation of silk fibroin microneedles, and summarizes the current research focus and future development trend in this field.

## 2. Structure and Properties of Silk Fibroin

### 2.1. Structure of Silk Fibroin

Silk is composed of silk sericin and silk fibroin [36]. Silk sericin is a kind of globulin with the characteristics of anti-oxidation, bacteriostasis and biocompatibility. Silk sericin can effectively reduce free radicals and reduce sun and air pollutants. Based on the above features, silk sericin is mainly used in medical cosmetology and other fields [37,38]. Silk fibroin is a kind of fibrous protein containing 18 types of amino acids, primarily glycine (Gly), alanine (Ala) and serine (Ser), accounting for more than 80% of the total content [39].

Silk fibroin protein is composed of three proteins, including a heavy chain (H chain), a light chain (L chain) and p25 glycoprotein, whose molecular weights are 390 kDa, 26 kDa and 28 kDa, respectively. In the silk fibroin secreted by the mature silkworm, the H and L chains are covalently connected by a pair of disulfide bonds [40]. An aggregate of six H-L chains and a p25 glycoprotein are assembled by hydrophobic action to form a basic unit, that is, H chain:L chain:p25 glycoprotein = 6:6:1 [41,42]. From the composition perspective, the H chain of silk fibroin is a highly ordered polymer formed by 11 hydrophilic regions through 12 hydrophobic regions [43] (Figure 1). Studies have found that the core part of the H chain is a β-folded structure formed by the orderly arrangement of the highly repeated sequence GAGAGX (X is mainly tyrosine or serine), which plays a significant role in the crystallization process of silk fibroin and is primarily distributed in the crystallization region [44,45]. In the amorphous region, there are essentially amorphous amino acids, such as tryptophan (Trp), phenylalanine (Phe) and tyrosine (Tyr), which have large side chain groups [46].

The secondary structure of proteins is mainly composed of the random coil, α-helix, β-sheet and β-turning [48,49]. Among them, the β-sheet includes the parallel β-sheet and antiparallel β-sheet. The latter structure is more stable in energy and produces shorter hydrogen bonds [50]. The random coil is unstable and will be transformed into the β-sheet under specific external forces, such as temperature, humidity, pH change and others. The β-sheet can generate physical crosslinking points under intermolecular force and hydrogen bonding, thus improving the mechanical properties of silk fibroin materials [25]. Amino acid repeating polypeptide fragments (GAGAGS) with minor or no side groups in the silk fibroin molecular chain are arranged in regular order to form hydrophobic β-sheet, which creates the possibility for further accumulation of hydrophobic β-sheet into silk II crystals [51,52]. Silk fibroin mainly contains two molecular conformations, Silk I and Silk II, among which Silk I is a β-turning conformation of II, and Silk II is an antiparallel β-sheet layered structure, which is stable and insoluble in water and most solvents, including strong acids and weak bases [53]. Silk I is easily transformed into Silk II under the influence of different solution conditions and temperatures, which some scientists have considered. Still, our recent study found that porous materials with the structure of Silk I show good long-term stability, thermal stability and chemical stability [54,55].

### 2.2. Biocompatibility of Silk Fibroin

One of the oldest natural polymers, silk has been successfully used for surgical sutures. Silk fibroin was purified by degumming. It was found that it was highly biocompatible and degraded slowly in vivo [56]. Since most of the amino acid residues in silk fibroin are non-polar, and the polar amino acid residues are in the crystallization region, the silk fibroin can be modified by chemical modification, and the modified silk fibroin has good blood compatibility [57].

Compared with other types of polymers, such as collagen and polylactic acid, silk fibroin-derived materials have better biocompatibility due to the β-sheet structure. Meinel [58] verified that the inflammatory response in vivo caused by silk fibroin film was equal to or smaller than that observed on collagen film and far smaller than that of polylactic acid film. TiO_2_ microspheres decorated with silk fibroin in situ showed enhanced biocompatibility compared with TiO_2_ microspheres [59].

### 2.3. The Advantages and Disadvantages of Silk Fibroin for Drug Release

In recent years, more and more researchers have found that silk fibroin has better characteristics for drug delivery than other biological materials, such as good biological stability of enzymes [26,27], antioxidant activity stability [28,29], biological stability in biomacromolecular drugs [30] and application in vaccines [31,32]. These unique advantages gradually improve the application prospect and irreplaceability of silk fibroin. Silk fibroin is often prepared into films, hydrogels, nanoparticles, microneedles and other forms according to its uses, which can be used in various fields such as medical cosmetology, tissue engineering, drug delivery and others.

The most common method of loading drugs with silk fibroin protein is to dissolve or blend the medicine directly into silk fibroin solution. This method is commonly used to prepare films [60] and hydrogels [61,62]. However, it is necessary to ensure that the structure and activity of the drug will not be affected during the manufacturing process. In addition, drugs can also be adsorbed or covalently crosslinked to the prepared silk fibroin matrix. Peroxidase can be covalently coupled to the silk fibroin scaffold using water-soluble carbodiimide [63]. Of course, whether a drug can be covalently crosslinked with silk fibroin or form a conjugate depends on the physical and chemical properties of the drug, and the use of this method is subject to certain restrictions [64]. Loading drugs into nanoparticles and microspheres is also a viable option. This enables targeted drug delivery and controlled drug release. Drug-loaded microspheres or nanoparticles can be blended with silk fibroin or coated on the surface of the silk matrix. However, the safety of nanoparticles is controversial, and they also face the challenges of low drug loading and wide size distribution.

Silk fibroin has a unique structure and can be adjusted to form different carrier forms. Among them, the secondary structure of silk fibroin protein has other characteristics and can be changed by external stimuli. The loading, release kinetics and stability of drugs can be processed by reversing the preparation and post-treatment methods of materials [65]. The loading efficiency and release curve of heparin-loaded silk fibroin nanofilm is controlled by the β-fold content after treatment with simple solvents (glycerol and methanol). The nanofilm can be used as a carrier to achieve continuous release of epirubicin [66]. The anion side group is increased by diazo coupling to modify the tyrosine side chain in silk fibroin, and the drug loading quantity is improved [67].

As an excellent biomaterial, silk fibroin has been prepared in many forms under mild processing conditions, including microspheres, films, hydrogels, liposomes, microneedles and microcapsules. However, these standard production techniques usually produce particles with a large distribution due to the irregular forces involved, resulting in low bioactivity of loaded enzymes, cells and growth factors [68]. At the same time, the degumming process not only reduces the mechanical strength of silk but also affects the structure, size, surface potential and drug release efficiency of microspheres in the process of silk fibroin purification due to the difference in molecular weight [69]. In addition, untreated silk fibroin microneedles are prone to fracture during insertion, and their rapid dissolution may lead to the sudden release of drugs, causing hypoglycemia and other side effects [70].

## 3. Types of Silk Fibroin Microneedles

According to the microneedle substrate and drug delivery mechanism, microneedles can be roughly divided into five different types [71] (Figure 2): (1) Solid microneedles can puncture the skin to improve drug transport rate and penetration rate, but their poke-patch delivery mode is inefficient, and it is difficult to control drug dosage [72,73]; (2) Drugs can be coated on microneedles on the surface of the needle tip. This coat-poke method often fails to meet the dose requirements and the dose is not easy to control [74,75,76,77]; (3) Hollow microneedles inject liquid drug preparation through internal holes. This poke-flow method overcomes the dose limitation, and the drug delivery velocity and dose can be controlled through free diffusion, external pressure or electronic control of external pressure. However, the structure of hollow microneedles is relatively complex. There are problems such as blockage, drug leakage, structural fragility and poor insertion [78,79,80]. (4) Dissolving microneedles are made of soluble polymers and encapsulated drugs. The disadvantages of hollow microneedles, such as high cost and a low dose of coating microneedles, are eliminated here. However, the method of poke-dissolve also has its disadvantages, including low mechanical strength and deposit of needle tip material in the cortex after dissolution [81,82,83]; (5) Swelling microneedles, also known as hydrogel microneedles, are a kind of microneedle that absorbs water and swells to form hydrogel after penetrating the skin, thus completing drug release. The drug release rate of the microneedles can be comparable to that of dissolved microneedles by regulating the change speed and degree of swelling. The tip material of this microneedle will not dissolve and deposit in the cortex, which solves the potential biocompatibility problem of dissolving microneedles and provides the possibility for the realization of a microneedle intelligent drug delivery system.

### 3.1. Dissolving Silk Fibroin Microneedles

Dissolving microneedles are used as a peridermal drug delivery system for drug and vaccine delivery [82], which not only avoids the shortcomings of conventional drug delivery methods but also uses biocompatible and biodegradable materials. This overcomes the problem of microneedles breaking in vivo. Recently, silk fibroin, as a natural biological material, has been used to manufacture dissolving microneedles due to its excellent biocompatibility and good mechanical properties [25,84].

The preparation method of dissolving silk fibroin microneedles is mainly mold casting, which is a method to prepare microneedles by mold casting and demolding [85]. As shown in the figure below, Cao et al. [86] described this method. First, the silk fibroin solution was mixed with the drug in a certain ratio, and then the mixed solution was evenly smeared on the polydimethylsiloxane (PDMS) microneedle mold. Finally, the PDMS mold filled with silk fibroin protein solution was sent to a constant temperature and humidity room for drying, and the microneedles could be detached after drying completely. Because the mold casting method is prepared at room temperature, the preparation conditions are mild, the cost is low, the shape is easy to control; thus, it is widely used in the preparation of soluble silk fibroin microneedles (Figure 3).

Although dissolving microneedles prepared with pure silk fibroin protein have many advantages, they are not accessible for piercing the skin in clinical application. In the actual operation process, there are different requirements for the tip and base of soluble microneedles. The end of the needle needs to be able to penetrate the cuticle, which requires sufficient strength, and the needle holder needs to be able to act as a platform to apply force. Silk fibroin is mechanically strong enough to be used as a needle tip. However, silk fibroin as a base is easily broken [82]. To solve this problem, Lau et al. [82] prepared a composite dissolving microneedle by using silk fibroin as the tip and polyvinyl alcohol (PVA) as the base. The process is shown in Figure 4a. Comparing the skin insertion conditions of pure silk fibroin microneedles and composite dissolved microneedles, the results showed that the composite microneedles could pierce the skin more effectively than pure silk fibroin microneedles because the PVA flexible base provided enough skin attachment and stress dispersion. In addition, Lin et al. [87] also proposed a method of preparing double-layer silk fibroin microneedles. They enhanced the mechanical strength of silk fibroin microneedles through glutaryl alcohol crosslinking and water vapor annealing. They then coated the enhanced microneedles with a mixture of silk fibroin and drug to obtain double-layer silk fibroin microneedles (Figure 4b). This method also facilitates skin piercing.

In addition to the mold casting method, Shin et al. [88] used riboflavin as a photoinitiator to prepare silk fibroin microneedles using digital light processing 3D printing technology. The mechanical properties of this method are good. A single needle can withstand the compression force of over 1000 mN and can successfully pierce pig skin. However, compared with the mold pouring method (Figure 5b), the microneedles prepared by 3D printing (Figure 5a) have poorer morphology and are limited by materials.

### 3.2. Swelling Silk Fibroin Microneedles

The drug release mechanism of swelling microneedles is as follows: when body fluid (the main component is water) penetrates the polymer skeleton, the molecular chains of polymer materials will stretch and expand [89,90]. By reducing the entanglement degree between polymers, the mesh size will be larger than the kinetic size of the drug, thus promoting drug release [91,92]. Its advantage is that the drug release rate can be adjusted by controlling the swelling degree of microneedles [30,35]. When harmful drug reactions or excessive drug release occurs, the microneedles can be controlled by timely removal [93].

Microneedles can be divided into two categories according to the solution: one is where the solution of the tip and the base is the same. Yin et al. [18] mixed small molecules with regenerated silk fibroin solution at different mass ratios, centrifuged repeatedly to ensure that the tip was filled entirely, and continued to add liquid and obtained uniform swelling of silk fibroin microneedles after air drying. Under the optimal ratio, the swelling degree of microneedles reached 650% (Figure 6a). Wang et al. [17] prepared silk fibroin swelling microneedles by adding proline to silk fibroin solution. The swelling tip was observed by an electron microscope. With the increase in proline dosage, the swelling degree decreased from 200% to 80%, and the solubility of the tip was less than 3%, which is an ideal release microneedle with swelling without dissolution. With this preparation method, microneedles with uniform drug distribution can be obtained. When the solution enters the microneedles, the overall structure will swell and the mesh size will become larger.

The other category is when the solution of the needle tip and the substrate are not the exact solutions, including but not limited to pure silk fibroin solution, functional silk fibroin solution and other polymer solutions. As shown in Figure 6b, Chen et al. [94,98] used the pre-gel of silk fibroin protein to form a tip with a semi-interpenetrating structure, with pure silk fibroin solution as the base. The internal phase presents the hydration phase, while the external phase presents the dehydration phase. The microneedles show channels under a hydration state under external stimulation. DeMuth and Boopathy et al. [32,99] used silk fibroin as the tip and polyacrylic acid as the base, as shown in Figure 6c. The drug was stored in the needle tip and annealed with methanol. The needle tip fell off in the body, and body fluids controlled the swelling and slow release. The slow swelling of the silk fibroin part from the tip and the complete release within a few days characterizes this preparation method. Moreover, silk fibroin has good biocompatibility and will not cause adverse reactions in the body.

In the above discussion, all PDMS molds are used, but there are differences in the solution configuration. Next, different molds and methods are discussed. One uses inkjet technology to precisely design various microneedles at other locations. As shown in Figure 6d, Wang et al. [33] filled the central 4 × 4 array with a drug solution, and the peripheral 4 × 2 displayed a functional silk fibroin solution. The peripheral microneedles would release quickly within a few minutes, while the inner peripheral drug solution would release slowly. The other method is hot drawing or using an inverted mold to make the master plate, pouring PDMS for a negative mold. Lee et al. [95] used a spatial discrete thermal stretching system to manufacture a microneedle template of poly (lactic acid-glycolic acid) copolymer with various geometric shapes and an inverted mold using PDMS (Figure 6e). The swelling fibroin microneedles prepared by this method can effectively pierce the skin and control drug release. In addition, high-speed micro-milling ([96], Figure 6f) and drilling on the wax plate ([97], Figure 6g) can also be used to prepare the positive mode of microneedles.

### 3.3. Intelligently Responsive Silk Fibroin Microneedles

Stimulation-responsive materials are generally considered intelligent; although they cannot perform advanced complex functions, they are the cornerstone of building advanced complex, intelligent systems [100]. Stimulus-responsive materials can change their size under specific stimulus conditions, which makes them widely used in drug delivery, disease diagnosis and substance detection [101,102]. Figure 7 shows the physical changes (such as expansion/contraction, assembly/dissociation, sol/gel, etc.), chemical changes (such as degradation, crosslinking, etc.) or a combination of the two after stimulation of the stimulation-responsive materials. These basic modules build the cornerstone of stimulus-responsive materials, and drugs can be integrated into the materials to respond to stimuli and test the intelligent delivery of drugs [103].

Dissolved microneedles and swelling microneedles can only release drugs spontaneously according to the established design procedures, which cannot meet the complex requirements of the human internal environment. Based on this, some researchers began to develop silk fibroin microneedles with the intelligent response for drug-controlled release. Compared with traditional drug delivery systems, smart drug delivery systems have more advantages: (1) the blood drug concentration is relatively stable; (2) drug efficacy is improved and drug dosage is reduced; (3) responsive delivery; (4) the number of drugs used is reduced [104,105]. Intelligent drug delivery refers to the responsive delivery of drugs to organisms by drug-carrying innovative response materials based on changes in physiological stimuli (pH value, temperature, enzymes and different biological molecules, etc.) or external stimuli (electric field, current, magnetic field, light and external mechanical force, etc.) [106,107]. The release mechanism based on physiological stimuli in organisms is primarily passive, which can respond to changes in the microenvironment in organisms to achieve intelligent drug delivery. Chen [94] reported a semi-interpenetrating network of hydrogel microneedles prepared from silk fibroin and phenylboric acid/acrylamide for glucose-responsive insulin delivery. 4-(2-acrylamide ethyl aminoformyl)-3-fluorophenylboric acid/N-isopropylacrylamide with a molar ratio of 7.5/92.5 was selected to synthesize phenylboric acid to ensure the highest glucose sensitivity. Hydrogels containing borate and silk fibroin with network structure were prepared by polymerization and catalysis of ammonium persulfate/tetramethylethylenediamine. A two-layer strategy was adopted to manufacture an intelligent microneedle composed of a semi-interpenetrating hydrogel containing silk fibroin and a needle body composed of silk fibroin, thus showing different swelling degrees under different glucose concentrations and achieving responsive insulin delivery. Due to the complexity of the physiological environment in vivo, it is often difficult to accurately control the release rate of such passive drug release, so active and targeted drug release mechanisms should be adopted to improve the overall therapeutic effect. Gao et al. [35] reported an intelligent origami silk fibroin microneedle dressing with innovative drug release, biochemical sensing, epidermal sensing, and wound healing monitoring capabilities. The microneedles were prepared by mixing silica colloidal solution with SF solution. After demolding, the microneedles were immersed in hydrofluoric acid to obtain the filament microneedles with inverse opal photonic crystal (IO PC) structure. The drug-laden N-isopropylacrylamide was poured into a microneedle with many pores and an IO PC structure. The volume phase transformation temperature of N-isopropylacrylamide is 37 °C, and when the temperature exceeds 37 °C, the microneedles produce volume shrinkage, so as to achieve drug release. Wang et al. [33] reported designing, manufacturing and applying a hetero-fibroin microneedle (SMN) patch to bypass the blood–brain barrier and deliver multiple drugs directly to the tumor site. SMN can induce rapid drug delivery by remote infrared triggering.

The above intelligent response of silk fibroin microneedles based on physiological or external stimuli has dramatically expanded the application of silk fibroin. However, the stimulation-responsive elements of these smart microneedles are not derived from the silk fibroin substrate; additional responsive features are added. Although introducing these stimulus-responsive elements brings more prosperous functions to the microneedles, it is also accompanied by many disadvantages, such as complex processing, low intelligent responsiveness and poor biological security.

Dissolving microneedles are made from a soluble polymer and encapsulated drugs. The method of poke-dissolve has some disadvantages, including poor mechanical properties and depositing of the needle in the cortex. At the same time, dissolving microneedles are not suitable for applications that require frequent dosing, such as insulin, which is the value of non-injectable dosing. In addition, microneedle dissolution allows the needle material to enter the dermis, which requires strict sterility throughout the process, resulting in higher costs for microneedle use, which negates its usefulness. Due to the rapid release of drugs, the dissolving microneedles have an effect similar to that of injection, which can cause a sudden rise in blood drug concentration in the body within a short period of time, which often leads to certain sequelae (such as hypoglycemia). For drugs with a short half-life, dissolving microneedles have difficulty maintaining effective blood concentration in vivo for a long period, which affects the therapeutic effect. Swelling microneedles, also known as hydrogel microneedles, are a kind of microneedle that absorbs water and swells to form a hydrogel after penetrating the skin, thus completing drug release. The microneedles can control the drug release rate by regulating the change speed and degree of swelling. The tip part of the microneedle will not be deposited in the cortex, which solves the potential biosafety hazard of dissolving microneedles, and also provides the possibility for the realization of microneedle intelligent drug delivery systems. The above two types of microneedles can only release drugs spontaneously according to established design procedures and cannot meet the complex needs of the human internal environment. Based on the basic platform of swelling microneedles, a microenvironment can be formed inside the swelling microneedles. Stimulation-responsive materials such as microgels and micelles in the microenvironment will undergo responsive intelligent changes under the stimulation of various external or internal physiological signals, especially structural changes (such as contraction, expansion and dissociation) or unique response pathways to control drug release.

## 4. Silk Fibroin Microneedles for Drug Delivery

The molecular weight is the most significant difference between biomacromolecule drugs and traditional small-molecule drugs [108,109]. In general, small molecule drugs refer to compounds with molecular weights less than 1000 Da, and most small molecule drugs have molecular weights less than 500 Da. However, biologic drugs are mostly proteins with huge molecular weights, usually more than 5000 Da, and complex structures.

### 4.1. Transdermal Delivery of Small Molecule Drugs

Temozolomide (TMZ), with a relative molecular mass of 194.15, is one of the drugs used to treat glioblastoma. It can be converted into the active product 5-(3-methyl-1-triazeno)imidazole-4-carboxamide, which promotes tumor cell apoptosis through the mismatched repair of methylation add-ons. Regarding the administration of TMZ, some researchers also adopted the method of percutaneous administration. Zhao et al. [110] loaded TMZ and paclitaxel with polyethylene glycol dimethacrylate (PEG-DMA) hydrogel to inhibit the growth of postoperative glioma. PEG-DMA gel is non-degradable and requires secondary surgical removal. By combining TMZ with other drugs, Wang et al. [33] prepared a microneedle patch using silk fibroin as material. The biosafety of silk fibroin prevents secondary clearance after surgery. Lee et al. [111] proposed a highly flexible microneedle device for vascular drug delivery. It comprises a highly porous silk fibroin membrane and silk fibroin tip. The microneedle device successfully wrapped around the outer surface of blood vessels and delivered the anti-proliferative drug Sirolimus to the injured vascular tissue. Melatonin is mostly an amine hormone used to regulate the sleep quality of people with insomnia. Currently, patients are often given oral medication, but the half-life of melatonin is very short, so it is difficult to maintain sufficient drug concentration in the body [112]. Qi et al. [34] used small molecular additives to change the silk fibroin protein’s crystalline structure and control melatonin’s release rate. In vitro drug release results showed that effective drug concentrations were quickly obtained during early administration. Melatonin is released continuously to maintain a stable concentration of the drug. The use of microneedles to deliver small molecule drugs not only supports the long-term stability of blood drug concentration in vivo but also avoids the liver first-pass effect caused by oral administration and improves the bioavailability of drugs.

### 4.2. Transdermal Delivery of Macromolecule Drugs

In terms of molecular weight, compared with traditional oral, subcutaneous injection and transdermal patches, microneedles can significantly improve the delivery of drugs with high-fat solubility, small effective dose and strong pharmacological effect, so as to achieve transdermal absorption of water-soluble drugs and biomacromolecules, especially for antigens or biomacromolecules [113,114]. The transdermal delivery of macromolecule drugs by silk fibroin microneedles is briefly described for the example of taking insulin for the treatment of diabetes.

Insulin, a protein drug with a simple spatial structure, comprises two peptide chains connected by disulfide bonds [115]. As a biological peptide, Insulin has poor stability and needs to be kept between 2–8 °C during distribution and storage [116].

The microneedles used for transdermal insulin delivery mainly include hollow microneedles [117,118], swelling microneedles [119] and dissolving microneedles [120,121]. The preparation process of hollow microneedles is complex and expensive [122], and the pinhole is easily blocked, thus affecting drug delivery [123]. In recent years, dissolving microneedles have attracted more and more attention due to their apparent advantages, such as simple preparation, high drug loading and one-step application, and they have been widely used for insulin delivery in studies. Silk fibroin has recently been used to manufacture dissolved microneedles due to its better mechanical properties, toughness and biocompatibility. Liu et al. [82] reported a multi-layer conical microneedle patch prepared from insulin-loaded pure silk fibroin solution as the outer layer and drug-free silk fibroin solution as the inner layer. The multi-layer microneedles showed good penetrating performance and hypoglycemic ability. However, untreated microneedles prepared with pure silk fibroin are prone to break and dissolve rapidly, which can lead to a burst of insulin release and hypoglycemia. Zhu et al. [30] reported a composite silk fibroin microneedle patch to solve this problem. The microneedle patch was prepared at 25 °C using a two-step microforming method. Unmixed silk fibroin was used for the dissolved tip, and the swelling backing was prepared using silk fibroin with proline modification. After the microneedle patch penetrates the skin, the tip is quickly dissolved by body fluids, and insulin is released. At the same time, body fluids contact the backing through microchannels formed by the tip of the needle, causing it to expand and subsequently release insulin continuously. Drug administration results in diabetic rats showed that blood glucose reached the lowest level within 3 h and recovered to the initial level 6 h later. The structure showed obvious hypoglycemic effects and could meet the need of rapid hypoglycemic reduction after meals. At the same time, continuous insulin release can be achieved without the risk of hypoglycemia. Insulin has good stability in this solid substrate, maintaining more than 90% of its biological activity after 30 days at room temperature. As a protein, silk fibroin can be hydrolyzed under the action of enzymes and thus eliminated from the body [124]. Cao et al. [86] selected silk fibroin modified by proline to prepare a swelling microneedle, whose microneedle structure was mainly Silk I crystalline structure. The drug loading of the microneedle patch is 20 IU/0.5 cm^2^, which can provide insulin sustained release for 12 h and control the blood glucose of diabetic rats to maintain normal levels, avoiding frequent administration.

### 4.3. Silk Fibroin Microneedles for Transdermal Administration of Vaccines

Vaccine administration by subcutaneous (SC) or intramuscular (IM) injection is the most commonly prescribed route for inoculation [125,126,127]. Vaccine activity and efficacy require strict requirements for vaccine distribution and transportation, which is also a significant challenge for developing countries [125,128]. Compared with IM/SC injection, transdermal vaccination with microneedles, as an emerging technology, could reduce the required amount of vaccines and further improve immunogenicity with the potential of lowering the cost of vaccination, expanding a promising administration route [129,130]. At the same time, commonly used vaccines are composed of antigens or antibodies, and their molecular weight is too large for transdermal delivery through traditional methods [131]. An interesting approach to dealing with this obstacle is the use of microneedles [125]. The acceptance of patients is improved, and the antigen/antibody activity can be stable in the dry environment, thus saving the cost of transportation and distribution.

Jordan et al. [128] evaluated the ability of microneedles prepared from silk fibroin to deliver vaccines against influenza, Shigella and Clostridium difficile infection. The microneedle carrier uses silk fibroin protein and has high biocompatibility. The stability and mechanical strength of the vaccine are also guaranteed. By inoculating the mice with separate antigens, it was found that a large amount of Shigella-protective invasion plasmid antigen (Ipa) was found in the mice, and no redness and swelling appeared on the skin where the microneedles were applied. The experiment proved that the microneedles had good biological safety and that the internal vaccine could be released smoothly. Archana et al. [32] focused on the study of stabilized recombinant HIV-1 envelope glycoprotein antibody variants. After the vaccine was mixed with silk solution, it was evenly poured into the PDMS mold and dried. Polyacrylic acid was used as microneedle base material. When applied to the skin, the polyacrylic acid backing will dissolve rapidly, and the silk fibroin tip containing the vaccine will be implanted into the epidermis/epithelium. The crystallinity of the silk fibroin will significantly improve the release rate of the vaccine. Blood epidermis/dermis was collected and analyzed 7, 14, and 21 days after the skin was punctured. Tip sequences containing labeled trimer were observed in the epidermis, and the total antigen signal decreased steadily within 14 days, which was longer lasting and more effective than the injection form. Peter et al. [99] also designed an implantable silk fibroin hydrogel composite microneedle with polyacrylic acid as the backing to obtain mixed microneedles, successfully achieving sustained skin vaccine release at a low level within 1–2 weeks. The implantable microneedle platform has the ability to optimize vaccine release dynamics, resulting in a more than 10-fold increase in antigen-specific T-cell and humoral immune responses compared to traditional parenteral needle immunization.

## 5. Clinical Transformation of Microneedles and Silk Fibroin Materials

### 5.1. Commercial Application of Silk Fibroin Medical Apparatus

Silk is a natural protein polymer that has been approved by the US Food and Drug Administration (FDA) for medical use. From a conversion perspective, to our knowledge, only a few silk fibroin-based medical products have received regulatory approval for clinical use worldwide to date (Table 1), including SERI scaffold with SILK VOIVE injectable implant (Sofregen Medical Inc., Framingham, MA, USA) and Silk Protein Wound Dressing (Suzhou Soho Biomaterial Science and Technology Co., Ltd., Suzhou, China). In 1995, AST-1 Silk Fibroin Wound Protective Dressing developed by Soochow University obtained a registration certificate, and clinical application began in 1996. On the basis of AST-1 Silk Fibroin Wound Protective Dressing, the product was adjusted according to clinical conditions, and Silk Protein Wound Dressing was launched and put on the market in 2012. Silk Protein Wound Dressing is indicated for second-degree burn wound healing. Detailed clinical data on Silk Protein Wound Dressing has not yet been published. In 2019, Zhejiang Xingyue Biotechnology Co., Ltd. (Yongkang, China) developed a silk fibroin regeneration and repair membrane. The full-layer skin defect experiment using New Zealand rabbits showed that the silk fibroin membrane could significantly reduce the wound healing time and improve the wound healing quality, which was superior to the control products on the market. In a 71-patient randomized controlled clinical trial (NCT01993030), the silk fibroin membrane significantly accelerated wound healing and reduced the incidence of adverse reactions compared with marketed controls. The SERI surgical scaffold was commercialized in 2013 for soft tissue support and repair. The current SERI is suitable for abdominal wall reconstruction and investigational plastic surgery applications, including total body molding, humeroplasty, abdominoplasty, mastexy, and breast reconstruction [132,133,134]. Some clinical reports indicate that SERI may cause side effects such as infection and poor stent integration in the later stage, requiring surgical removal. Silk Voice, developed by Sofregen, is the only approved natural silk protein injection. Silk Voice was approved in 2018 for the treatment of vocal cord mediation and vocal cord insufficiency.

### 5.2. Clinical Application of Silk Fibroin Medical Apparatus

Over the past 20 years, there has been a resurgence of interest in silk for biomedical use, which has led to many clinical trials. Table 2 shows the clinical trials of silk fibroin medical devices. Nowadays, silk fibroin is widely used in wound repair dressings and orthopedic repair materials. The primary application forms are dressing, film, stents, etc.

Silk fibroin materials show excellent application prospects in skin wound healing due to their hemostatic properties, low inflammatory potential, air permeability and anti-bacterial ability. Silk Protein Wound Dressing is a silk fibroin sponge-silicone double-layer scaffold for skin wound healing and has received regulatory approval worldwide for clinical use. However, this silk fibroin-based medical product is not widely used in clinical trials today. Zhang et al. [135] compared the wound healing ability of Silk Protein Wound Dressing in a preclinical animal model and then conducted a randomized, single-blind phase I clinical trial. The silk fibroin film was dried for 100 min at 65 °C and RH90%. It is a waterproof film with a thickness of 64.9 μm. In a rabbit model of full-layer wound healing, wounds treated with silk fibroin film showed better epidermal remodeling and granulation tissue than those treated with Silk Protein Wound Dressing. The clinical trial enrolled 71 patients (36 randomly assigned to the membrane group and 35 randomly assigned to the Silk Protein Wound Dressing group). The wound healing rate of patients in the silk fibroin film group was significantly faster than that in the Silk Protein Wound Dressing group, and 100% wound healing was achieved in the silk fibroin film group by day 14 after injury. As the wound heals, the silk fibroin film spontaneously detaches from the regenerated skin area. The exact mechanism of the improved clinical performance in the silk fibroin film group compared to the Silk Protein Wound Dressing group is unclear.

This study demonstrated the ability to manufacture silk fibroin membranes as wound dressings for successful skin repair and regeneration under good manufacturing practices. The potential for relatively simple modifications to silk membranes for additional functions, such as binding pores or the introduction of bioactive molecules [136], makes silk membranes particularly attractive as wound dressings.

### 5.3. Clinical Application of Microneedle Devices

This paper searched and sorted the clinical applications of silk fibroin medical devices and microneedle devices of other materials [137]. We listed completed and ongoing clinical trials based on information provided at www.clinicaltrials.gov. In clinical trials, researchers mainly verified the clinical safety and efficacy of microneedles for drug delivery systems, as shown in Table 3. These clinical data indicate that microneedles for drug delivery are effective quickly and with less insertion pain. This mainly includes dissolving microneedles, hollow microneedles, coated microneedles and microdetectors used for administering vaccines, macromolecular drugs and monitoring human physiological conditions.

## 6. Future Development Trends

This article reviewed the characteristics of silk fibroin and the preparation methods of different microneedle types. More importantly, the development of silk fibroin microneedles for drug delivery is introduced in detail.

Currently, most of the drugs on the market are small-molecule drugs prepared by chemical synthesis. However, the number of new drugs such as proteins, peptides and antibodies is on the rise. The market is estimated to exceed US $217 billion, accounting for 10% of the pharmaceutical market [138]. These biomacromolecules differ from chemically synthesized small molecules and have a relatively large molecular size and conformational flexibility. Peptides and proteins are considered multi-domain biopolymers, consisting of residues with rich cation, polarity and hydrophobicity differences [139]. Unlike many materials, silk fibroin is inherently stable to changes in temperature and humidity. The unique block copolymerization structure enables it to assemble into a nanometer crystalline domain. As a result, silk fibroin has better characteristics in drug delivery than other biological materials, such as good biological stability of enzymes, antioxidant activity, and biological stability for biomolecular drugs and vaccines. These unique advantages gradually improve the application prospect and position of silk fibroin [27,140,141,142]. Therefore, silk fibroin is an excellent candidate material for preparing microneedles. Silk fibroin microneedles have developed from soluble microneedles and swelling microneedles to intelligent responsive microneedles, which can effectively carry small molecule drugs, large molecule drugs and vaccines. Silk fibroin microneedles have achieved the same or even more powerful effects as needle injection in terms of form and function. Despite these encouraging results, more efforts are needed to accelerate the entry of silk fibroin microneedles into the medical market.

The first issue is safety. Although the biosafety of silk fibroin has widely been recognized, basic studies remain necessary, including those involving cytotoxicity, acute/chronic systemic toxicity, hemolysis testing, stimulation testing, implantation testing, implant testing, intradermal reaction testing, biodegradation testing, carcinogenic testing, etc. The safety of silk fibroin microneedles still needs further thorough verification in different animal or preclinical models.

The second is the drug-carrying dose and type. Although some active biological enzymes (horseradish peroxidase, glucose oxidase [143]) and some vaccines (influenza vaccine, HIV vaccine, adenovirus vaccine [144] and others) have been successfully used, it is still worth studying how to increase the carrying dose, which will provide effective dose support for the transformation from experimental animals to human beings. At the same time, about 40% of research drugs are hydrophobic, among which there are many macromolecular drugs. Further study is required to improve the carrying dose and conform to the water-soluble environmental processing of silk fibroin microneedles. Of course, this must be carried out according to the specific drug. Another possible solution is to extend the current microneedle manufacturing method. The existing template method has shortcomings in the large-scale preparation of microneedles and the delivery of large doses of drugs. The extended microneedle manufacturing method is expected to solve this problem.

The third is intelligence. Traditional microneedle drug delivery can only spontaneously slow-release drugs according to pre-set procedures. With the development of personalized medicine, the traditional way cannot meet the needs of the complex environment in the human body. The silk fibroin microneedles with stimulatory response function are expected to play a significant role in the drug-controlled release. This can help improve the effectiveness of treatment and reduce the risk of inappropriate drug use. Intelligent responsive silk fibroin microneedles can be realized in two ways: one is to chemically modify the microneedle substrate to obtain stimulation-responsive silk fibroin material; the other is to add stimulation-responsive elements (such as stimulation-responsive nanoparticles, microspheres, vesicles or supramolecular aggregates) to pure silk fibroin protein. This will help the silk fibroin microneedles achieve intelligent responsiveness.

The current regulatory approval process based on the microneedle patch is not ideal, mainly due to the advanced technology. For future market applications, standardized regulation of sterilization methods, durability, safety and disposal of microneedles after use is also required. The ultimate sterilization of microneedle patches can save most of the cost compared to aseptic manufacturing. Several microneedle-based sterilizations have been studied. According to the requirements of the European Pharmacopoeia, the sterilization of microneedle patches is based on steam sterilization, dry heat sterilization and ionizing radiation. According to the different properties of microneedle substrates, the sterilization process has different effects. Wet and dry heat sterilization can damage the morphology and penetration ability of silk fibroin microneedles, but gamma irradiation sterilization seems to be the only viable option. After gamma ray sterilization, the activity and release of drugs contained in microneedles did not change significantly [145,146].

In addition, there is insufficient data on the side effects of microneedles (e.g., skin irritation, microneedle substrate deposition), and more research is needed to select polymers that minimize skin irritation. Silk fibroin protein has incomparable advantages in biosecurity. However, it is still necessary to determine the specific amount of silk fibroin remaining in the skin after removal of microneedles and the removal of residues in the later stage. Deposition of the substrate may not be an important problem in the case of a single microneedle administration but may be significant if the microneedles are used frequently over a long period of time.

## 7. Conclusions

This review focuses on the critical research progress of silk fibroin microneedles since inception, analyzes in detail the structure and properties of silk fibroin, types of silk fibroin microneedles, drug delivery application and clinical transformation progress, and summarizes the future development trend in this field. It also proposes future research directions for silk fibroin microneedles in increasing drug loading dose and enriching drug loading types as well as exploring silk fibroin microneedles with stimulation-responsive drug release functions. The safety and effectiveness of silk fibroin microneedles should be adequately verified in clinical trials at different stages. What we have achieved determines where we stand, and what we will do in the future determines how far we can proceed.

## Figures and Tables

**Figure 1 pharmaceutics-15-00355-f001:**
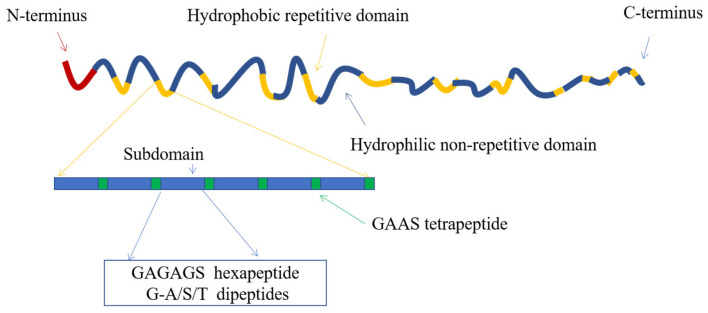
Diagram of H chain of silk fibroin. Image used with permission of [47].

**Figure 2 pharmaceutics-15-00355-f002:**
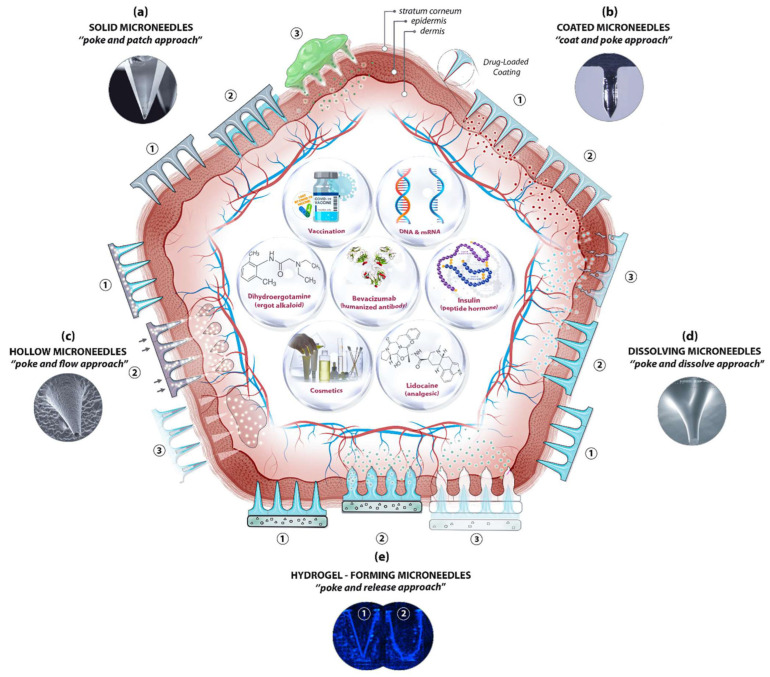
Schematic diagram of drug delivery method based on microneedles (MN): (**a**) solid MN; (**b**) coated MN; (**c**) hollow MN; (**d**) dissolving MN; (**e**) swelling MN. The step-by-step process for each delivery method is numbered from 1 to 3. It shows representative microscopic images of MN types and examples of deliverable substances, such as insulin, vaccines, DNA and RNA. Image used with permission of [72].

**Figure 3 pharmaceutics-15-00355-f003:**
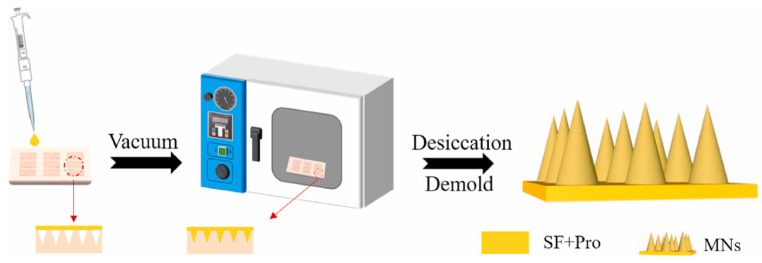
Preparation of microneedles by template pouring. Image used with permission of [86].

**Figure 4 pharmaceutics-15-00355-f004:**
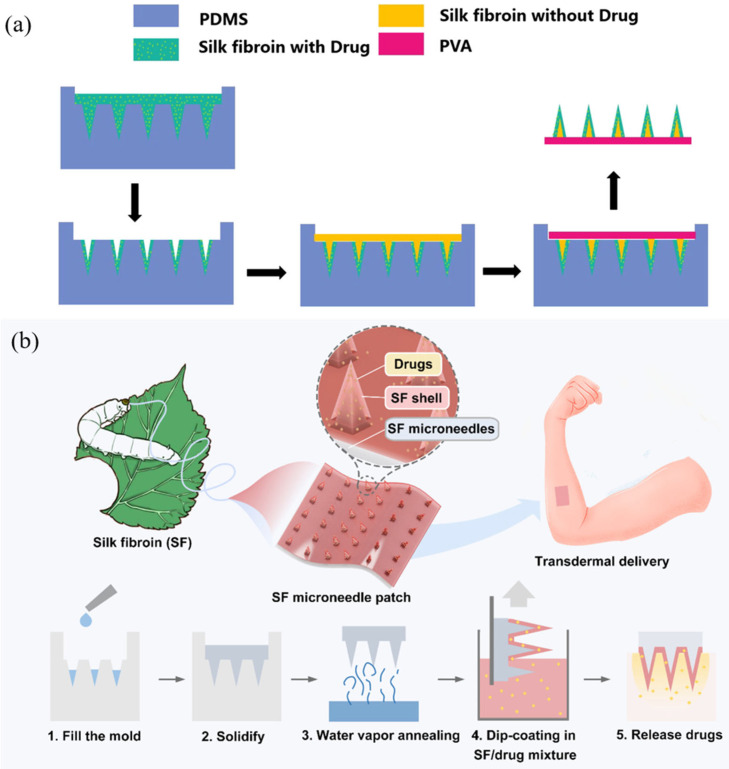
(**a**) Schematic diagram of composite dissolved microneedles: drug-loaded silk fibroin protein is used as the tip of the outermost layer, drug-free silk fibroin protein as the second layer, and PVA as the base; (**b**) Schematic diagram of the production of double-layer silk fibroin microneedles: the silk fibroin solution containing glutaraldehyde is poured into the PDMS mold, and after drying, steam annealing treatment takes place, and the silk fibroin solution is soaked in medicine on the surface of the dry microneedles. Image used with permission of [82,87].

**Figure 5 pharmaceutics-15-00355-f005:**
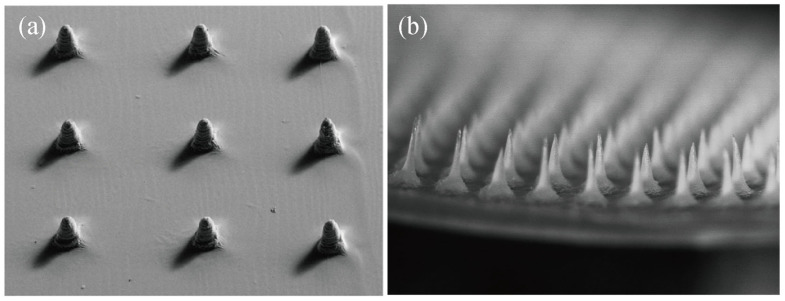
Morphologies of microneedles prepared by different methods: (**a**) 3D printing; (**b**) Formwork pouring method. Image used with permission of [30,88].

**Figure 6 pharmaceutics-15-00355-f006:**
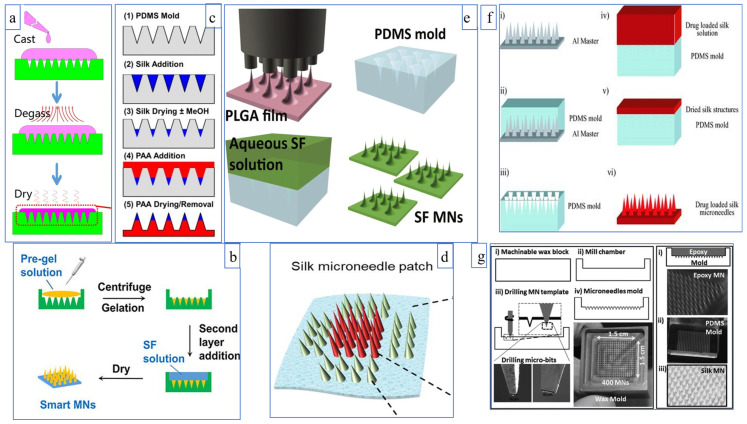
(**a**) The tip and the base are the same solution; (**b**,**c**) Different solution between tip and base; (**d**) Different microneedles in different positions; (**e**) Negative mold for hot-drawn material; (**f**) Aluminum needle master plates prepared by high-speed micro-milling; (**g**) Wax plate drilling for template preparation. Image used with permission of [18,32,33,94,95,96,97].

**Figure 7 pharmaceutics-15-00355-f007:**
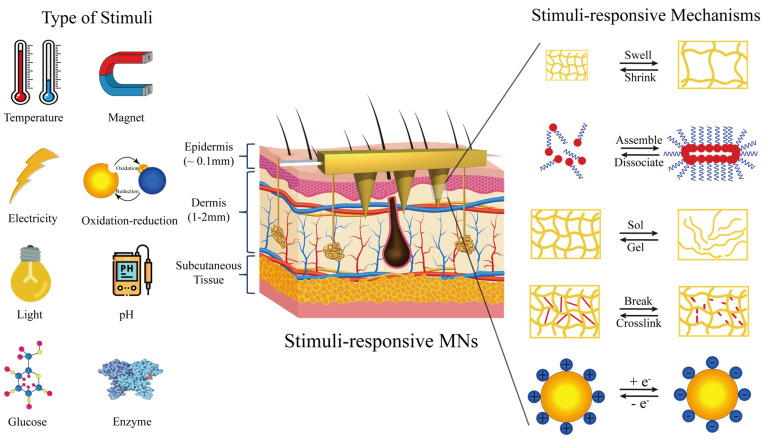
Types of stimulus response.

**Table 1 pharmaceutics-15-00355-t001:** Commercially available silk fibroin products.

Official Title	Type	Company	Application	Announced Date	Registration Certificate Number
AST-1 Silk Fibroin Wound Protective Dressing	Single-layer dressing	Soochow University (Suzhou Institute of Silk Technology)	It is used to prevent bacterial infection of the wound. It presents no irritation to the skin, can promote the growth of wound cells, and has good permeable steam.	1995	Su1995-95264098
Silk Protein Wound Dressing	Double-layer dressing	Suzhou Soho Biomaterial Science and Technology Co., Ltd.	It is used for clinical second-degree wound healing; is suitable for mild and severe acne, skin allergies and laser photon treatment of early postoperative pigmentation; and reduces the formation of fatigue marks.	2012	Su2012-2640182
SILK VOICE^®^ Injectable Implant	Injectable implant	SOFREGEN MEDICAL Inc.	It is used for clinical second-degree wound healing; is suitable for mild and severe acne, skin allergies and laser photon treatment of early postoperative pigmentation; and reduces the formation of fatigue marks.	2012	-
Silk fibroin membrane dressing	Single-layer dressing	Zhejiang Xingyue Biotechnology Co., Ltd.	It covers the skin wound, blocks the external bacteria, and prevents the granulation tissue from growing into the dressing. At the same time, the water in the wound blood and exudate is discharged as water vapor to provide a healing environment for the wound, and the blood cells and other visible components left behind form a scab.	2020	SFDA2020-3140593
SERI^®^ Surgical Scaffold	Scaffold	SOFREGEN MEDICAL Inc.	A temporary brace used for soft tissue support and repair to strengthen defects where weaknesses or gaps exist. This includes soft tissue reinforcement in plastic and reconstructive surgery, as well as soft tissue reconstruction in general.	2021	-

**Table 2 pharmaceutics-15-00355-t002:** Clinical trials of silk fibroin medical devices.

Official Title	Type	Condition	Primary Purpose	Status	Start Date /Completion Date	Trial ID
A Multi-Center Open Study to Evaluate the SeriACL™ Device for Primary Anterior Cruciate Ligament Repair	ACL Reconstruction (SeriACL™ Device)	Phase 1	Treatment	Unknown	2007.6/ 2008.10	NCT00490594
Randomized, Active-controlled, Single-blind, Parallel Two-group Trial of HQ^®^ Matrix Medical Wound Dressing and Sidaiyi^®^ Wound Dressing for the Treatment of Donor Site Wounds	HQ^®^ Matrix Medical Wound Dressing	Not Applicable	Treatment	Completed	2013.8/ 2015.6	NCT01993030 CQZ1800141
Efficacy and Safety of Wound Dressing Containing Silk Fibroin with Bioactive Coating Layer Versus Medicated Paraffin Gauze Dressing in the Treatment of Split-thickness Skin Graft Donor Sites	Silk fibroin with bioactive coating layer dressing	Phase 1/2	Treatment	Completed	2014.3/ 2015.5	NCT02091076
A Prospective Open-Label Study to Evaluate the Safety of the Meniscal Repair Scaffold, FibroFix™ Meniscus, in the Treatment of Meniscal Defects	FibroFix™ Meniscus scaffold	Not Applicable	Treatment	Terminated (Safety devices explanted. 12 m post-explant safety f/u as agreed with UK MHRA)	2015.4/ 2017.10	NCT02205645
Multi-center, Randomized, Active-controlled, Single-blind, Parallel Two-group Trial of HQ^®^ Matrix Soft Tissue Mesh and ULTRAPRO^®^ Partially Absorbable Lightweight Mesh for the Treatment of Inguinal Hernia	HQ^®^ Matrix Soft Tissue Mesh	Not Applicable	Treatment	Unknown	2015.7/ 2016.12 (estimated)	NCT02487628
Comparison of Microbial Adherence to Various Sutures in Patients Undergoing Oral Surgery	Silk suture	Not Applicable	Treatment	Unknown	2016.1/ 2017.1 (estimated)	NCT02653924
A Pilot Feasibility Randomized Controlled Trial to Assess the Clinical and Cost Effectiveness of Dialkylcarbamoylchloride (DACC)-Coated Postoperative Dressings Versus Standard Care in the Prevention of Surgical Site Infection in Clean or Clean-contaminated Vascular and Cardiothoracic Surgery	DACC-Coated Post-Operative Dressing	Not Applicable	Treatment	Recruiting	2017.1/ 2025.1 (estimated)	NCT02992951
Silk Scaffold Surgical Incision Dressing Interventional Study	Experimental silk/adhesive prototype	Phase 1	Assessment	Recruiting	2022.8/ 2023.5 (estimated)	NCT05508945
Safety and efficacy of absorbable silk fibroin film for alveolar ridge preservation after extraction	Absorbable silk fibroin film	Phase 3/4	Assessment	Completed	-	CQZ1900597 ChiCTR-IOR-17025031

**Table 3 pharmaceutics-15-00355-t003:** Clinical research progress in microneedle devices.

Official Title	Type	Phase	Primary Purpose	Status	Start Date /Completion Date	Trial ID
Insulin Delivery Using Microneedles in Type 1 Diabetes	Hollow	Phase 2/3	Treatment	Completed	2009.2/ 2014.1	NCT00837512
Clinical Assessment of a Novel Microprobe Array Continuous Glucose Monitor for Type 1 Diabetes	Microprobe glucose sensor	Phase 1/2/3/4	Diagnostic	Completed	2013.11/ 2018.6	NCT01908530
A Phase I Study of the Safety, Reactogenicity, Acceptability and Immunogenicity of Inactivated Influenza Vaccine Delivered by Microneedle Patch or by Hypodermic Needle	Dissolving microneedle	Phase 1	Prevention	Completed	2015.5/ 2019.7	NCT02438423
Microneedle Sensing of Beta-lactam Antibiotic Concentrations in Human Interstitial Fluid	Biosensors	Phase 1	Device Feasibility	Completed	2019.2/ 2020.12	NCT03847610
A clinical trial of dose-response using a microneedle array containing Japanese encephalitis vaccine in healthy adult individuals	Dissolving microneedle	Phase 1/2	Prevention	Completed	2019.7/ 2020.2	jRCTs011190004
A Phase I/II, Double-blind, Randomized, Active-controlled, Age De-escalation Trial to Assess the safety and Immunogenicity of a Measles Rubella Vaccine (MRV) Microneedle Patch (MRV-MNP) in Adults, MRV-primed Toddlers, and MRV-naïve Infants	Dissolving microneedle	Phase 1/2	Assessment	Recruiting	2021.5/ 2022.6 (estimated)	NCT04394689

## Data Availability

Not applicable.

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
