# Peer review of "Silk Fibroin Microneedles for Transdermal Drug Delivery: Where Do We Stand and How Far Can We Proceed?"

_pharmaceutics, 2023, doi:10.3390/pharmaceutics15020355_

Round 1
Reviewer 1 Report
In this manuscript, the authors present an in-depth, comprehensive review of silk microneedle based research so far. To make it more relevant, it may be worthwhile to include an idea of costs (with material and manufacture of silk MN) and how it compares against other materials of construction and processes. Also, since the MN devices will need to be sterile (or not, if the authors argue that), the ways of sterilizing silk based devices (or prepare them in a clean environment+additional cost of it) should be commented on. Lastly, while the authors touch up upon regulatory approval of silk based products, it would be useful to add the regulatory approval processes of MNs and how silk based MN will fare in that regard.
Author Response
We appreciate the reviewer’s positive evaluation of our work. Thanks very much for taking your time to review this manuscript. We really appreciate all your generous comments and suggestions! Please find my revisions in the re-submitted files.
Reviewer 2 Report
This review focuses on silk fibroin microneedles. The authors described the structure of silk fibroin and the various fabrication methods for making microneedles, Although there are several excellent publications in this research area, this is a significant addition to the literature, and I recommend the manuscript for publication.
Comments: The abstract is not well-written, and this should be revised. For example, ‘Microneedles are considered a new transdermal method, which is improved by combining a transdermal patch with a hypodermic needle’, ‘clinical transformation progress.
Author Response
We sincerely thank the reviewers for taking the time to review our manuscript and for your recognition of our review. We have rewritten the abstract and corrected the inappropriate statements.
Reviewer 3 Report
Manuscript titled " Silk Fibroin Microneedles for Transdermal Drug Delivery: Where Do We Stand and How Far Can We Proceed? " reviews recent progress on the research of silk fibroin microneedles for drug delivery. Structure and properties of silk fibroin, types of silk fibroin microneedles, transdermal delivery of different molecule drugs, clinical application of silk fibroin-based biomaterials and their challenges in future are discussed in detail. It is interesting because microneedles for drug delivery and human health monitoring is very helpful for disease manage and monitoring and has the great potential for clinical application. However, the current manuscript suffers from several major issues (detailed below). With further clarification or improvement, the reviewer believes this study can be considered for publication.
1. There are plenty of grammar mistakes in the manuscript, please check the sentences carefully within the entire manuscript.
Line 31, ‘maintaininges’should be ‘maintaining’;
Line 33, ‘m2’should be ‘m2’;
Line 131, ‘…degumming.I It …’should be ‘…degumming. It …’;
Line 222 ‘…properties[25,86].’should be ‘…properties [25,86].’;
2. A comparison among different types (absorbable, swelling and smart) of microneedles for drug delivery is needed to the best of readers.
3. The future and challenges of the microneedle-based drug delivery platforms are not summarized quite well. Microneedle consists of at least two categories: controllable biomaterials production and scalable manufacturing approaches. Hence it is better to discuss the current scenario, challenges and future from at least these two perspectives.
4. For all the tables, the authors are suggested to check the format required for publication in the journal. Clear paragraph should be presented in all the tables.
5. The figures presented in this manuscript are very poor. In Figure 6, each subfigures should be carefully cited. In Figure 7, the figure is so simple without enough information for this broad and hot research fields.
6. For review papers, all the figures modified from the previously published papers should be with copyright permission.
7. The keywords should be more specific. For instance, the words ‘Microneedles’and ‘Biocompatibility’are too general.
8. The abstract should be further condensed.
Author Response
We are grateful to the esteemed Reviewer for providing the comments and suggestions. We have tried our best to answer all the issues raised and updated the revised manuscript accordingly.
- There are plenty of grammar mistakes in the manuscript, please check the sentences carefully within the entire manuscript.
Line 31, ‘maintaininges’should be ‘maintaining’;
Line 33, ‘m2’should be ‘m2’;
Line 131, ‘…degumming.I It …’should be ‘…degumming. It …’;
Line 222 ‘…properties[25,86].’should be ‘…properties [25,86].’;
Thank you for your careful review. We are very sorry for the mistakes in this manuscript and inconvenience they caused in your reading. We modified these errors at the appropriate places in the article.
- A comparison among different types (absorbable, swelling and smart) of microneedles for drug delivery is needed to the best of readers.
Thank you very much for your reminding. We have added the comparison of dissolved microneedles, swelling microneedles and smart microneedles in Section 3.
Dissolving microneedles are made from a soluble polymer and encapsulated drugs. The method of poke-dissolve has some disadvantages, including poor mechanical property and deposit of needle in the cortex. At the same time, dissolving microneedles are not suitable for applications that require frequent dosing, such as insulin, which is the value of non-injectable dosing. In addition, microneedle dissolution allows the needle material to enter the dermis, which requires strict sterility throughout the process, resulting in higher costs for microneedle use, which negates its usefulness. Due to the rapid release of drugs, the dissolving microneedles have the effect similar to that of injection, which can cause a sudden rise in blood drug concentration in the body within a short period of time, which often leads to certain sequelae (such as hypoglycemia). For drugs with short half-life, dissolving microneedles are difficult to maintain effective blood concentration in vivo for a long time, which affects the therapeutic effect. Swelling microneedles, also known as hydrogel microneedles, are a kind of microneedles that absorb water and swell to form hydrogel after being penetrated into the skin, thus completing drug release. The microneedles can control the drug release rate by regulating the change speed and degree of swelling degree. The tip part of the microneedle will not be deposited in the cortex, which solves the potential biosafety hazard of dissolving microneedle, and also provides the possibility for the realization of microneedle intelligent drug delivery system. The above two types of microneedles can only release drugs spontaneously according to established design procedures, which cannot meet the complex needs of human internal environment. Based on the basic platform of swelling microneedles, a microenvironment can be formed inside the swelling microneedles. Stimulation-responsive materials such as microgels and micelles in the microenvironment will undergo responsive intelligent changes under the stimulation of various external or internal physiological signals, especially structural changes (such as contraction, expansion and dissociation) or unique response path-ways to control drug release.
- The future and challenges of the microneedle-based drug delivery platforms are not summarized quite well. Microneedle consists of at least two categories: controllable biomaterials production and scalable manufacturing approaches. Hence it is better to discuss the current scenario, challenges and future from at least these two perspectives.
Thank you very much for your advice. We have already added a discussion of these two categories in Section 6.
Another possible solution is to extend the current microneedles manufacturing method. The existing template method has shortcomings in the large-scale preparation of microneedles and the delivery of large doses of drugs. The extended microneedle manufacturing method is expected to solve this problem.
The third is intelligence. Traditional microneedle drug delivery can only spontaneously slow-release drugs according to pre-set procedures. With the development of personalized medicine, the traditional way cannot meet the needs of the complex environment in the human body. The silk fibroin microneedles with stimulatory response function are expected to play a significant role in the drug-controlled release. This can help improve the effectiveness of treatment and reduce the risk of inappropriate drug use. Intelligent responsive silk fibroin microneedles can be realized in two ways: one is to chemically modify the microneedle substrate to obtain stimulation-responsive silk fibroin material; the other is to add stimulation-responsive elements (such as stimulation-responsive nanoparticles, microspheres, vesicles or supramolecular aggregates) to pure silk fibroin protein. This will help the silk fibroin microneedles achieve intelligent responsiveness.
The current regulatory approval process based on microneedle patch is not perfect, mainly due to the advanced technology. For future market applications, standardized regulation of sterilization methods, durability, safety and disposal of microneedles after use is also required. The ultimate sterilization of microneedle patches can save most of the cost compared to aseptic manufacturing. Several microneedle-based sterilizations have been published. According to the requirements of the European Pharmacopoeia, the sterilization of microneedle patches is based on steam sterilization, dry heat sterilization and ionizing radiation. According to the different properties of microneedles substrate, the sterilization process has different effects. Wet and dry heat sterilization can damage the morphology and penetration ability of silk fibroin microneedles, but gamma irradiation sterilization seems to be the only viable option. After gamma ray sterilization, the activity and release of drugs contained in microneedles did not change significantly [148, 149].
In addition, there is insufficient data on the side effects of microneedles (e.g. skin irritation, microneedle substrate deposition), and more research is needed to select polymers that minimize skin irritation. Silk fibroin protein has incomparable ad-vantages in biosecurity. However, it is still necessary to determine the specific amount of silk fibroin remaining in the skin after removal of microneedles and the removal of residues in the later stage. Deposition of the substrate may not be an important problem in the case of a single microneedle administration, but may be significant if the microneedles are used frequently over a long period of time.
- For all the tables, the authors are suggested to check the format required for publication in the journal. Clear paragraph should be presented in all the tables.
Thank you for your careful review. We are very sorry for the mistakes in this manuscript and inconvenience they caused in your reading. We have modified the tables format according to the requirements of the journal.
- The figures presented in this manuscript are very poor. In Figure 6, each subfigures should be carefully cited. In Figure 7, the figure is so simple without enough information for this broad and hot research fields.
Thank you very much for your comments. Each of the subfigures in Figure 6 has been carefully cited and obtained copyright permission. We have added the mechanism of stimuli-responsive to Figure 7, hoping that the readers will get a better understanding of it.
Figure 7. Types of stimulus-response
- For review papers, all the figures modified from the previously published papers should be with copyright permission.
Thank you very much for your reminding. The references cited in all figures have all obtained copyright permission.
- The keywords should be more specific. For instance, the words ‘Microneedles’ and ‘Biocompatibility’ are too general.
Thank you very much for your comments. We've reworked the keywords to make them more specific.
Keywords: Silk fibroin microneedles; Transdermal delivery; Intelligently responsive; Clinical transformation
- The abstract should be further condensed.
We sincerely thank the reviewers for taking the time to review our manuscript. We have rewritten the abstract and corrected the inappropriate statements.
Abstract: Microneedles are a patient-friendly technique for delivering drugs to the site of action in place of traditional oral and injectable administration. Silk fibroin represent interesting polymeric bio-material because of their mechanical properties, thermal stability, biocompatibility, and possibility of control via genetic engineering. This review focuses on the critical research progress of silk fibroin microneedles since its birth, analyzes in detail the structure and properties of silk fibroin, types of silk fibroin microneedles, drug delivery application and clinical trial, and summarizes the future development trend in this field. It also proposes the future research direction of silk fibroin microneedles increasing drug loading dose and enriching drug loading types and exploring silk fibroin microneedles with stimulation-responsive drug release function. The safety and effectiveness of silk fibroin microneedles should be further verified in clinical trials at different stages.

Round 2
Reviewer 3 Report
Authors have addressed all my comments. I have no further comments.